# Assessing the Ability of Luojia 1-01 Imagery to Detect Feeble Nighttime Lights

**DOI:** 10.3390/s19173708

**Published:** 2019-08-26

**Authors:** Xue Li, Zhumei Liu, Xiaolin Chen, Jie Sun

**Affiliations:** 1Key Laboratory of Earthquake Geodesy, Institute of Seismology, China Earthquake Administration, NO. 40 Hongshance Road, Wuhan 430071, China; 2School of Geography and Information Engineering, China University of Geosciences, Wuhan 430074, China

**Keywords:** Luojia1-01 satellite, feeble nighttime light, cutoff threshold, spatial correspondence, area consistency, image composition, background noise

## Abstract

The Luojia1-01 (LJ1-01) satellite launched on 2 June 2018 provides a new option for nighttime light (NTL) application research. In this paper, four types of human settlements, such as cities, counties, towns and villages, are sampled to evaluate the potential of LJ1-01 to detect feeble NTL by comparing with the NTL images from the Defense Meteorological Satellite Program’s Operational Linescan System (DMSP/OLS) and the Visible Infrared Imaging Radiometer Suite (VIIRS) on the Suomi National Polar-Orbiting Partnership Satellite. First, the landscape indices and cutoff threshold method are applied to enhance signal-noise ratio (SNR). Then, the detection accuracy of samples is evaluated to determine the optimal cutoff threshold for each NTL data source. After that, the spatial correspondence of different NTL images and the area consistency between the samples and NTL footprints are compared. Finally, after the discussion of feeble NTL detection and the influence of clouds, moonlight and image composites, it can be concluded that LJ1-01 is more suitable for detection feeble NTL objects, while great importance should be attached to the measures to eliminate the noise in LJ1-01 image and make LJ1-01 more widely used: (1) In the study area, a suitable cutoff threshold of LJ1-01 image can be set to 0.1 nano-Wcm^−2^sr^−1^, which is lower than that of VIIRS image (0.3 nano-Wcm^−2^sr^−1^), and this enables LJ1-01 to reserve more information of NTL, especially the feeble NTL. Moreover, the minimum area that can be identified by NTL footprints from LJ1-01 is 0.02 km^2^, while that of VIIRS and DMSP are 0.3 km^2^ and 4.5 km^2^, respectively. (2) The cutoff threshold method can identify the range of NTL with more noise, but cannot eliminate the noise separately. The filtering method and the image composition method may play more important role in the applications of LJ1-01 data.

## 1. Introduction

In recent years, nighttime light (NTL) has been a hot topic in remote sensing research. Because of the unique ability of NTL to reveal human activities as well as NTL’s advantages of spatiotemporal continuity and independence [1,2], NTL is widely used to estimate gross domestic product (GDP) [3,4,5], population [6,7] and electricity consumption [8,9], urbanization monitoring [10,11,12,13,14], ecological environmental changes [15], and major event assessment [16,17,18,19,20].

Taking urbanization monitoring as an example, due to the brighter NTL emitted by cities, current research on NTL mainly focuses on the high-intensity NTL [21,22,23,24,25]. However, the low-intensity NTL are rarely analyzed. Sometimes, these low-intensity NTL is even ignored by using an empirical thresholding technique [26,27]. This is acceptable when the research object is city, because low-intensity NTL is often not recognized as part of the city. Since the low-intensity NTL sources are more susceptible to be interfered by system and random noise, discarding low-intensity NTL can reduce errors and uncertainty in the urbanization analysis.

However, low-intensity NTL is not worthless. It represents a low level of human settlement, such as towns and villages, or some irregular human activities, which may be caused by traffic or the construction of a building or a road. For example, scholars have found that low-intensity NTL can be used to explain the consistent between the maintenance of the main roads after the 2008 Wen Chuan earthquake and NTL strips on the time series NTL images from DMSP [18]. In this paper, these low-intensity NTL emitted by small size of human settlements and unstable human activities are defined as “Feeble NTL”. Feeble NTL can play an important role in settlement surveys in mountainous areas and in post-disaster emergency assessments [19,20]. NTL image with higher sensitivity and SNR is useful for the studies of human activity-related monitoring, especially in sparsely populated areas. In this regard, the Luojia 1-01 (LJ1-01) NTL Satellite may be able to offer help.

The LJ1-01 NTL Satellite, which was successfully launched by China on 2 June 2018, was jointly developed by Wuhan University and related institutions to test and verify new NTL remote-sensing technologies. The satellite has a high spatial resolution of 130 m and a high radiometric quantization (14 bits) with a swath of 250 km. In theory, LJ1-01 can draw a global NTL image within 15 days. The data of LJ1-01 has been used in many kinds of researches includes urban mapping [27], estimating socio-economic parameters [28], urban community housing price assessment [29], impervious surface detection [30] and investigating artificial light pollution [31]. However, the research on the low-intensity NTL of LJ 1-01 is still blank.

The objective of this study is to assess the ability of LJ1-01 to detect feeble NTL. After comparing with the NTL images from the Visible Infrared Imaging Radiometer Suite (VIIRS) on the Suomi National Polar-Orbiting Partnership (NPP) Satellite, and the Defense Meteorological Satellite Program’s Operational Linescan System (DMSP/OLS), the advantages and disadvantages of LJ1-01 are concluded, and some suggestions for improvement are proposed.

## 2. Materials and Methods

### 2.1. Study Areas

In order to evaluate the ability of LJ1-01 to detect the feeble NTL, a suitable study area is essential. First of all, we cannot choose NTL images from economically developed urban areas, because the NTL of these places is often bright. Secondly, it is better not to select images from the plains, as the high population density in these places often leads to an excessive NTL brightness. Both factors considered, the key monitoring area around the Three Gorges Reservoir in China was selected as the study area. The Three Gorges Reservoir is located at the junction of China’s second and third elevation steps, where average elevation rises from less than 500 m to 1000–2000 m. This reservoir is one of the largest water conservation projects in the world, with a total storage capacity of 39.3 billion cubic meters. The total area of artificial lakes created by the Three Gorges Dam is 1084 square kilometers, and the project forced more than 1 million residents to relocate. The study area, which is shown in Figure 1, is located along the Yangtze River from Yichang City to Badong County. Due to the inconvenient transportation, the local economy is underdeveloped, even lower than the national average level. Most of the study area is mountainous, therefore the local population density is low. According to the China Population Spatial Distribution Kilometer-grid Dataset in 2015 [32], the minimum population density of the study area is 67 people/km^2^, the maximum density is 9682 people/km^2^, and the average density is 197 people/km^2^, which is much lower than the average population density in the plains of China (359 people/km^2^).

### 2.2. Data Source

In this paper, images from DMSP/OLS and NPP/VIIRS, which are commonly used in NTL research [33], are compared with LJ1-01 imagery to test feeble NTL detection ability. The parameters of the various NTL satellites are shown in Table 1.

The daily NTL images of LJ1-01 are freely available for researchers, and they can be downloaded from the High-Resolution Earth Observation System of the Hubei Data and Application Center. In order to cover the entire study area, two images acquired on 18 August 2018 and 3 September 2018 are obtained from the center [34]. The VIIRS Day/Night Band data are monthly composites that are filtered to exclude data impacted by stray light, lightning, lunar illumination, and cloud coverage. The composite image of the study area in September 2018 is downloaded from the Earth Observations Group (EOG) at the National Oceanic and Atmospheric Administration (NOAA)/National Centers for Environmental Information (NCEI) [35]. DMSP/OLS data are cloud-free annual composites. Since they are only updated to 2013, the F182013 composite image is downloaded from the EOG at NOAA/NCEI [36]. The reason why these three different products with different spatial resolution and temporal coverage are compared is that only daily data of LJ 1-01 is available recently but no composites such as monthly or annual data. On the other hand, both monthly composites of VIIRS and annual composites of DMSP are the most accessible NTL data and widely used in the current NTL researches. Although both VIIRS images and DMSP images have their respective suitable uses, this article mainly uses them to compare the detection of feeble NTL with LJ1-01 images. In addition, high-resolution Google Map images are also used for survey verification in this study.

### 2.3. Nighttime Light Imagery Processing

High-precision geometric positioning of NTL images is the basis of the comparison. First, after clipping and mosaicing, the LJ1-01, VIIRS, and DMSP NTL images of the study area are re-projected to a WGS 1984 UTM Zone 49 N projection. Then a nearest-neighbor algorithm is applied to resample the VIIRS and DMSP images to the same resolution as that of the LJ1-01 image (i.e., 130 m). 

The positioning accuracy of LJ1-01 NTL images without geometric control points (GCPs) is less than 650 m [37], which is almost five times the resolution of the image. Therefore, the LJ1-01 images are geometrically corrected using the Google Map images before analysis. Because of the high spatial resolution of LJ1-01, the road network can be identified as the GCPs. Figure 2 shows a comparison between various NTL and Google Map images.

Since the DMSP NTL images are not corrected onboard, the digital number (DN) is used to represent relative brightness. In the contrast, images from both VIIRS and LJ1-01 are absolute radiation-corrected. The radiance conversion formula of the LJ1-01 image [38] was:(1)$$L=DN^{3/2}\cdot 10^{-10}$$
where *L* is the radiance value after absolute radiation correction in Wm^−2^sr^−1^μm^−1^ and DN is the gray value of the image. It is observed that the radiation unit of VIIRS is nano-Wcm^−2^sr^−1^, which is not consistent with that of LJ1-01. The reason for this fact is that the radiance of LJ1-01 is converted to the central wavelength, while that of VIIRS uses the full-band radiance. Therefore, the radiation unit of LJ1-01 is converted to that of VIIRS by the following formula:(2)$$L^{\prime }=10^{5}\cdot \Delta \lambda \cdot L$$
where *L’* is the radiation value after unit conversion and Δ*λ* is the bandwidth of LJ1-01. Figure 3 shows processed images of the study area. It should be noted that the study area is located at the junction of the two LJ1-01 images; therefore the number of NTL footprints in the overlapping region appears to be more than that in anywhere on the mosaic image, and this is the real reason why there is a high density vertical strip on the mosaic image of LJ 1-01 (shown as Figure 3c). 

### 2.4. Detection Capability Assessment

In order to evaluate the ability of different NTL sources to detect feeble NTL, three steps are adopted. First, the relationship between NTL footprints and cutoff thresholds are compared. Secondly, according to the samples, the producer’s accuracy and user’s accuracy of the NTL data for different cutoff thresholds are evaluated. Finally, the effects of weak NTL are compared under the optimal cutoff threshold. 

The weak NTL received by the sensors is easily interfered with by system and background noise. The range of NTL values that are susceptible to noise can be roughly identified by the histogram. With the help of landscape indices [39], the spatial distribution of the NTL’s footprints for different cutoff thresholds can be analyzed. Although these landscape indices are mainly invented for ecological service analysis, such as analysis of specific land cover and land use types [40,41], it is considered that there is a high correlation between land use types and the intensity of surface human activities [42]. As we know, there is also a correlation between NTL and human activities. Therefore, the NTL’s footprints are regarded as a special landscape in this study. Through the landscape indices, the spatial characteristic of the NTL patches’ distribution can be quantitatively evaluated under different cutoff thresholds to indentify the proportion of feeble NTL and noise. Table 2 shows the landscape indices used in this study.

According to the different scales of human settlements in the study area, human activities are divided into four levels: city-level, county-level, town-level, and village-level. Samples at each level are randomly selected from the National Geodatabase [43]. Since the human settlements in the database are point-vector data, it is necessary to manually extract the urbanized area of each sample according to the high-resolution images from Google Map. Then the producer’s accuracy (PA) and user’s accuracy (UA) of different NTL data sources for different cutoff thresholds are calculated as:(3)$$PA=N_{co}/N_{as}$$
(4)$$UA=N_{co}/NP$$
where *N_co_* indicates the number of samples that are checked out by the NTL footprints, *N_as_* indicates the total number of samples and *NP* indicates the total number of patches of NTL’s footprints.

The optimal cutoff threshold for different NTL data is determined separately. The spatial correspondence, area consistency, and NTL intensity are compared between the NTL footprints and the samples. The spatial correspondence indicates the separability of the NTL footprints and the samples (shown in Figure 4). Good separation helps to improve the accuracy of object identification. The area consistency between the NTL footprints and the samples indicates the diffusivity of NTL. The higher the consistency, the smaller is the NTL spread. NTL intensity can reveal the level of human activity.

## 3. Results

### 3.1. Cutoff Threshold and Noise

Since the unit of the pixel value of the DMSP image is inconsistent with that of LJ 1-01 and VIIRS images, moreover, the original DMSP image of the study area has no pixels with a DN value smaller than 4. This indicates that the annual composite has been cut off by the threshold. Therefore, only the cutoff thresholds for VIIRS and LJ1-01 NTL images are analyzed in this section.

The histogram in Figure 5 shows that the NTL intensity of most pixels in the VIIRS image is concentrated near 0 and 0.2. When the NTL intensity of pixels greater than 0.3, the number of pixels gradually stabilize and then slowly decline. Considering that the background value of the VIIRS image is slightly greater than 0, and this has been shown in Figure 3b, it indicates that the pixels of VIIRS image with NTL intensity ranging from 0 to 0.3 may be more susceptible to noise. The histogram of the LJ1-01 images shows a decline trend. As seen, the histogram of the LJ1-01 image pixels with NTL intensity greater than 0.1, is very similar to that of the VIIRS image pixels with NTL intensity greater than 0.3. This similarity indicates that both two NTL images in their respective intervals are less affected by noise. On the other hand, it also means that the dissimilar curves (while the NTL intensity of LJ 1-01image pixels smaller than 0.1 and the NTL intensity of VIIRS image pixels smaller than 0.3) may be caused by noise.

In most urbanization studies, a cutoff threshold method is often applied for NTL images to facilitate extraction of the urban area and also to reduce the error caused by system and random noise. As mentioned above, it is difficult to distinguish feeble NTL from noise. However, with the help of landscape indices, the proportion of noise in all NTL information can be identified. If there were no noise, when the cutoff threshold increased from 0 to 1 (nano-Wcm^−2^sr^−1^), the NP of NTL patches would reduce linearly, and the CA of NTL patches would be gradually stabilized. A similar situation can be observed in the PD and MPS. However, since the NTL intensity caused by noise is often low, the size of individual noise patch is often small. 

If the cutoff threshold is set too small, NP and CA will increase significantly, and the trends of PD and MPS will also be different from before. Therefore, the inflection of these indices’ curves will help to determine the optimal cutoff threshold to minimize the effects of noise. Figure 6 shows the changes in landscape indices as the cutoff threshold rise. Based on the above reasoning, the optimal threshold for LJ1-01 should be greater than 0.1, while the optimal threshold for VIIRS should be greater than 0.3 or 0.4.

### 3.2. Sample Detection

Four types of human settlement samples are used to evaluate the feeble NTL detection capability with different NTL data sources. The categories and quantities of the samples are one city, three counties, 47 towns, and 248 villages. It must be realized that the sizes of urbanized area of different human settlement levels may be variable around the world. In the study area, the size of urbanized area of the city-level samples is greater than 150 km^2^. The average sizes of urbanized area of the county-level and town-level samples are 26.7 km^2^ and 0.87 km^2^ respectively. Then, the average size of urbanized area of the village-level samples is smaller than 0.1 km^2^. The degree of matching between the NTL patches and the samples is calculated, and Figure 7 shows the detection accuracy of different NTL data with different cutoff thresholds.

The NTL from cities and counties is usually very bright, and it can be easily observed. However, the focus of this article is on feeble NTL. That is the reason why the number of samples for town and villages are much larger than that of cities and counties. When calculating accuracy, the city-level samples and county-level samples are counted in combination with town-level samples. Figure 7a shows that the DMSP images could not identify the village samples. However, the producer’s accuracy of DMSP images for the town and larger settlement samples is 76.5%. The producer’s accuracy of VIIRS and LJ1-01 images gradually decrease as the cutoff threshold increased. For samples at town-level and above, the producer’s accuracy of the LJ1-01 images is higher than that of the DMSP images when the cutoff threshold is less than 0.8. The same situation occurres with the VIIRS images when the cutoff threshold is less than 0.6. When the cutoff threshold of the LJ1-01 images is in the range from 0.2 to 0.6, the producer’s accuracy of LJ1-01 is similar to that of VIIRS with a cutoff threshold between 0.4 and 0.5. For village-level samples, the DMSP images cannot identify any village samples. The producer’s accuracy of VIIRS and LJ1-01 NTL images for village samples is very low, especially when the VIIRS cutoff threshold is greater than 0.4 and the LJ1-01 cutoff threshold is greater than 0.1. This indicates that DMSP images can only detect NTL of human settlements above the town-level. VIIRS and LJ1-01 NTL images could identify NTL from a small number of village-level human settlements, but the NTL of these settlements is so weak that it is very close to the systematic error and is easily confused.

Figure 7b shows the user’s accuracy of different NTL data for human settlement samples. Twenty seven of 31 patches in the DMSP images are detected by the samples, and the relevant user’s accuracy is 87.1%, which is the highest among the NTL data sources. The patches in the VIIRS images are more fragmented. As the cutoff threshold rise, the user’s accuracy of the VIIRS images gradually stabilizes at around 0.25. The patches in the LJ1-01 images are the most fragmented, and therefore LJ1-01 have the lowest user’s accuracy. The reason may be that the LJ1-01 images are made up of daily data, and the effects of systematic and random noise are greater than on monthly average images. To reduce the noise, patches in the LJ1-01 images that are smaller than four pixels are eliminated. Then the user’s accuracy of the treated LJ1-01 images is nearly the same as that of the VIIRS images.

### 3.3. Detection Assessment

Too many NTL patches (caused by a low cutoff threshold) mean too much noise interference. However, an over-high cutoff threshold will filter out too much weak NTL information. Therefore, an optimal cutoff threshold for different NTL data sources should be identified for subsequent analysis. Unfortunately, it is difficult to accurately determine the optimal cutoff threshold using a quantitative approach. However, by analyzing Figure 6 and Figure 7, the range of the optimal threshold can be roughly determined by manual. Figure 6c,d suggest that the optimal threshold for LJ1-01 should be greater than 0.1, while the optimal threshold for VIIRS should be greater than 0.3 or 0.4. Figure 7a illustrates that the producer’s accuracy are relatively stable while the LJ1-01 cutoff threshold is between 0.2 and 0.6, and the VIIRS cutoff threshold is between 0.4 and 0.5. Figure 7b shows that when the LJ1-01 cutoff threshold is greater than 0.3 and the VIIRS cutoff threshold is greater than 0.4, the user’s accuracy is relatively stable, but there is not much difference from the previous one. According to the above analysis, the optimal cutoff threshold for VIIRS was set to 0.4 and that for LJ1-01 to 0.2. 

Due to the poor outcome of village-level sample detection, this paper only compares the spatial correspondence of NTL footprints and samples at town-level and above. One sample to one footprint (OSTOF) and one sample to multiple footprints (OSTMF) indicate that NTL patches have good separation, while multiple samples to one footprint (MSTOF) represent poor separation. Figure 8 shows the spatial correspondence of different NTL data sources under the optimal cutoff threshold. 44 of 51 samples with town-level and above could be separately identified using LJ1-01 images; only one sample is undistinguished. The spatial correspondence of the VIIRS and DMSP images is inferior to that of the LJ1-01 images. This can be attributed to the high spatial resolution of the LJ1-01 images, which generated a small overglow around the NTL emitted by human settlements. The overglow on the NTL image is usually appeared as a NTL patch, which size is larger than that of the real light source. Overglow may be caused by optical diffraction of the imaging lens and atmospheric scattering. A smaller overglow makes the target more distinguishable.

Figure 9a shows the area statistics of NTL patches identified by the town-level samples. The area of the NTL patches from the LJ1-01 images is relatively stable, but the area of the NTL patches from the DMSP images varies greatly. 

Comparing the area of the town-level samples with the area of the corresponding NTL footprints (see Figure 9b), the area of the corresponding NTL footprints from the LJ1-01 samples has the highest consistency with the area of the town-level samples. The consistency of the DMSP samples is the worst. This indicates that LJ1-01 is more accurate in recognizing targets.

Figure 10 shows the NTL intensity statistics for the patches, where DMSP counted by the DN value and VIIRS and LJ1-01 counted by the radiation value. The footprints of the DMSP images could identify only city, county and most town-level samples. The difference in the total and mean value of the NTL at different levels is obvious. The footprints of the VIIRS and LJ1-01 images could also identify most samples above the town-level. In addition, there are many patches on both images where the distributions of total NTL intensity and mean value are very similar to the town-level patches.

## 4. Discussion

### 4.1. Feeble NTL Detection

The town-level and above samples could be detected by all three NTL images from DMSP, VIIRS, and LJ1-01. The difference is that the DMSP images could only detect part of samples at town-level and above, and the minimum area of NTL patches in DMSP image is 4.5 km^2^. However, the VIIRS and LJ1-01 images could detect most of these level samples and some village-level samples, which could not be detected by the DMSP images, and the minimum area of NTL patches in VIIRS and LJ1-01 images are 0.3 km^2^ and 0.1 km^2^ separately. Moreover, the minimum area of villege-level NTL patches detected by LJ1-01 is 0.02 km^2^. In addition, because of the high resolution, the overglow on the LJ1-01 images is the minimum. It means that the size of the NTL patches on the LJ1-01 image is closer to the real object area.

Except for the DMSP images, the number of NTL’s footprints on the VIIRS and LJ1-01 images is far greater than the number of samples. Some of these footprints represent human settlements that are not selected as samples, and some are noise, while others may be caused by unstable human activities such as traffic or industry. Two linear feeble NTL strips are separately visible on the north and south of the study area on VIIRS image (see Figure 3b). The distributions of the two strips are consistent with the specific highways (highways number G50 and G42). However, this phenomenon is not observed on the LJ1-01 image.

The non-sample patches on DMSP image are mainly linear in shape with low DN values. Considering that the distribution of these patches is consistent with the distribution of some roads (Figure 11a–c), they may be caused by traffic or some temporary engineering activities, such as roads maintenance. The non-sample patches with higher total sum of NTL on the VIIRS image are related to industrial activities (Figure 11d,e) and traffic such as highway G50 (Figure 11f). The non-sample patches with both higher total sum of NTL and NTL mean on the LJ1-01 image include industrial activities (Figure 11g), road traffic (Figure 11h,j), as well as some villages (Figure 11i,k,l).

### 4.2. Clouds and Moon Phases

Comparing with the monthly and annual composites, daily NTL images from LJ1-01 exhibit many more fragmented patches. Some of these represent industrial activities or small human settlements such as villages, but more of them represent background noise. The size and number of NTL patches caused by noise are mainly determined by moonlight and cloud reflection. Brighter moonlight and more cloud will cause more noise. The intensity of moonlight can be estimated from the phase of the moon [44], and the amount of cloud can be obtained through the website of the National Meteorological Information Center [45]. The NTL intensity at the time of the LJ1-01 images in the study area is almost the same; the two scenes are observed at slightly less than one-half of full moon. Therefore, in this study, the main factor causing noise is the amount of cloud. After mosaicking the cloud map of the study area on 22 August 2018 and 3 September 2018 from 22:00 to 23:00, a map of the amount of cloud in the study area is generated and is shown in Figure 12a. The cloud percentages in most areas are between 20% and 60%. Figure 12b shows the linear relationship between the amount of cloud and the PD of the NTL patches in the LJ1-01 images in the study area. The correlation coefficient indicates that the amount of cloud is significantly positively correlated with the number of NTL patches that are considered to be noise. However, relying on daily data, it is difficult to eliminate the effects of noise.

### 4.3. Image Composition and Sensor Parameters

Image composition, resolution, and satellite overpass time are also key factors affecting sensor detection of feeble NTL. First, the image composition method is considered as the most important of these factors. Figure 13 shows an image coverage map of DMSP and VIIRS in the study area. Every pixel of the DMSP image is composed of from 50 to 68 scenes of daily data, while the pixels of the VIIRS image are based on 0 to 9 scenes. The greater the number of scenes composite, the less the noise in the image. However, it also blocks much the feeble NTL. This may be the reason why the DMSP images could identify only human settlements above the town-level, but the VIIRS images could show more human activities. At present, LJ1-01 provides only daily data, which contain not only the NTL emitted by human activities but also large amounts of system and background noise. An appropriate image composition method can help to filter out noise while preserving real human activities.

Secondly, resolution of NTL image is another very important factor. There is no doubt that the ground resolution of LJ1-01 images is higher than the DMSP and VIIRS images. This may play an important role in a smaller overglow, which is helpful for identifying more detailed spatial distribution of human activities. Higher resolution makes the area of patches closer to the area of the actual target object. Better spatial correspondence makes it possible to identify human activities at smaller scale and to obtain more detailed information on the structure of human settlements.

Last but not least, the overpass time of the NTL satellites also play an important role in feeble NTL detection. The overpass time of DMSP satellite is about 19:30 local time, while that of VIIRS satellite is about 1:30 local time. At 19:30, it is usually the most frequent time for human activities in a city at night, such as social intercourse, entertainment, and physical exercise. At this time, the city’s NTL tends to be more and brighter. It is a good data source to extract the urbanized area. However, the bright NTL of the city will produce a large overflow, which will cover most information of feeble NTL around the city. Moreover, sunlight may not have entirely disappeared at this time in some places especially in the western part of China in summer, and it may cause more noise. VIIRS images are suitable for urbanization extraction and socioeconomic parameter estimation. The overpass time of VIIRS makes the feeble NTL more significant than the background, however, some useful information of NTL will be lost, because some human activities have stopped and cannot be recorded by sensors at that time. The overpass time of LJ1-01 is about 22:30 local time, which makes it possible to record more NTL emitted by human activities with less interference by the Sun. Therefore, in theory, the LJ1-01 NTL images are more valuable in feeble NTL detection.

## 5. Conclusions

In this study, a series of experiments are carried out to analyze the ability of different NTL images to detect feeble NTL in a selected study area. After comparing the experimental results with the DMSP and the VIIRS images, the following conclusions are drawn for the LJ1-01 images:LJ1-01 NTL imagery has a better potential for feeble NTL detection. For example, in the study area, the minimum cutoff threshold for LJ1-01 image is 0.1 (nanoWcm^−2^sr^−1^), while that of VIIRS image is 0.3 (nanoWcm^−2^sr^−1^). This allows LJ1-01 images containing more useful information of NTL, especially the feeble NTL. In addition, with the optimal cutoff threshold, the minimum area of town-level NTL patches that can be identified from LJ1-01 image is 0.1 km^2^, while that of VIIRS and DMSP images are 0.3 km^2^ and 4.5 km^2^, respectively. Moreover, the minimum area of village-level NTL patches which are detected by LJ1-01 is 0.02 km^2^. It enables LJ1-01 to detect more small size NTL patches caused by human activities, and this is considered to be useful when analyzing the spatial structure of human settlements. Besides in the aspect of feeble NTL detection, the overpass time of the LJ1-01 also makes its data more valuable.A method of reducing the noise from LJ 1-01 imagery should be proposed, especially to reduce the noise caused by the cloud reflection. The cutoff threshold method is not the best way, because it cannot distinguish the noise from feeble NTL but discards the parts of NTL containing more noise. Perhaps, the filtering method and the image composition method can play a greater role in noise reduction. However, in the image composition, how to distinguish the noise from the temporary human activities, such as industrial activities and road traffic, and to retain the effective NTL is also a problem that has to be settled.

## Figures and Tables

**Figure 1 sensors-19-03708-f001:**
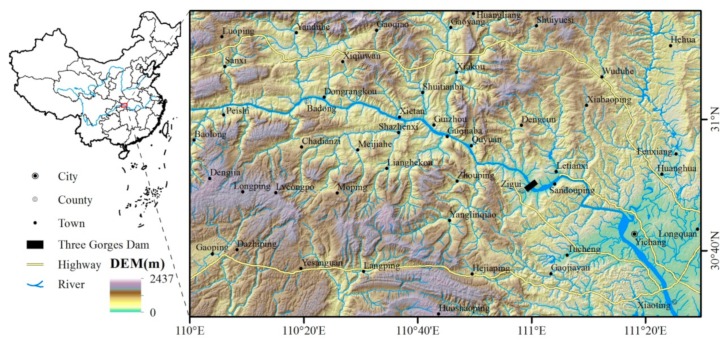
Topographic map of the study area.

**Figure 2 sensors-19-03708-f002:**
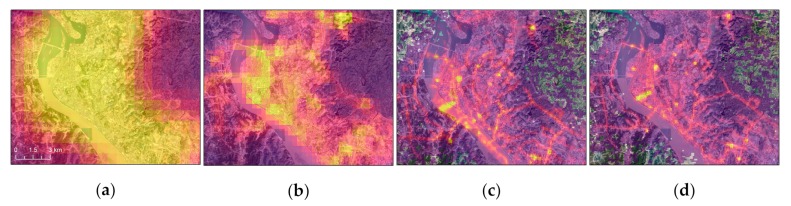
Geolocation performance of DMSP, VIIRS, and LJ1-01 NTL imagery: (**a**) overlay of DMSP and Google Map images for Yichang City; (**b**) overlay of VIIRS and Google Map images for Yichang City; (**c**) overlay of LJ1-01 and Google Map images for Yichang City; (**d**) overlay of corrected LJ1-01 and Google Map images of Yichang City.

**Figure 3 sensors-19-03708-f003:**
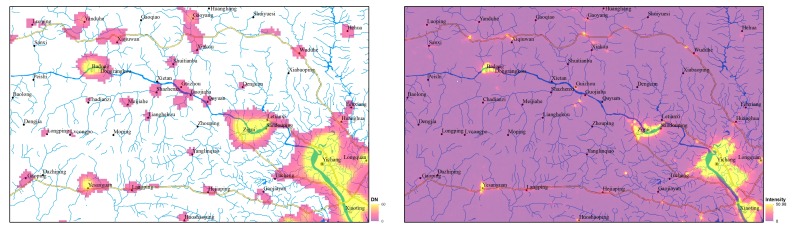
Sources of study area data: (**a**) DMSP NTL image; (**b**) VIIRS NTL image; (**c**) LJ1-01 NTL image; (**d**) Google Map image. When the NTL intensity or the DN value of the pixel is 0, the pixel is set to be transparent.

**Figure 4 sensors-19-03708-f004:**
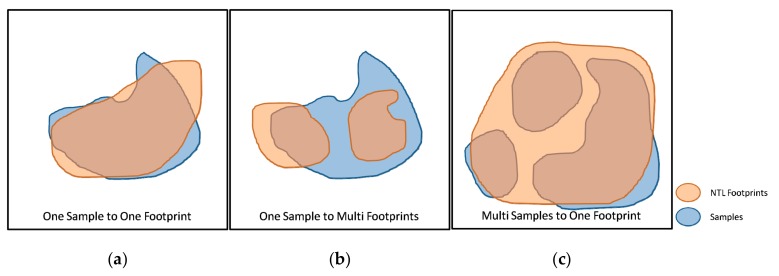
Spatial correspondence of NTL footprints and samples: (**a**) One sample to one footprint; (**b**) One sample to multi footprints; (**c**) Multi samples to one footprint.

**Figure 5 sensors-19-03708-f005:**
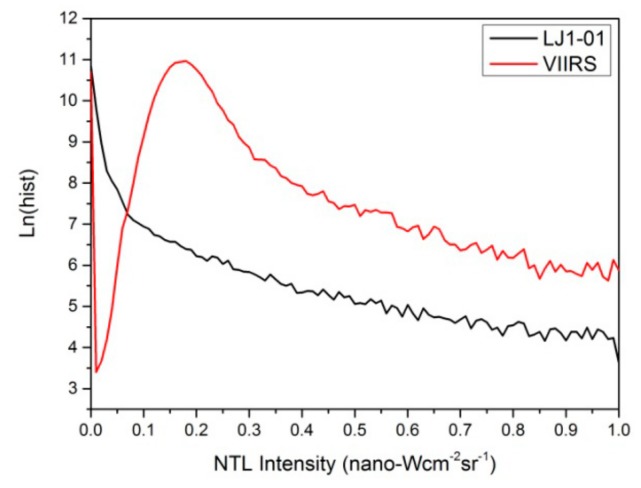
Histogram of VIIRS and LJ1-01 NTL images. The y-axis represents the logarithm of the number of pixels.

**Figure 6 sensors-19-03708-f006:**
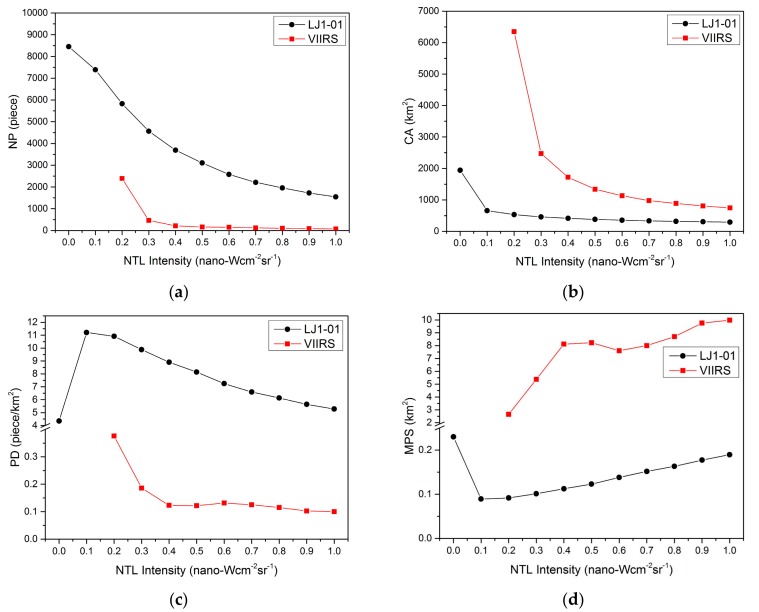
Landscape indices of footprints in different VIIRS and LJ1-01 NTL images: (**a**) NP; (**b**) CA; (**c**) PD; (**d**) MPS.

**Figure 7 sensors-19-03708-f007:**
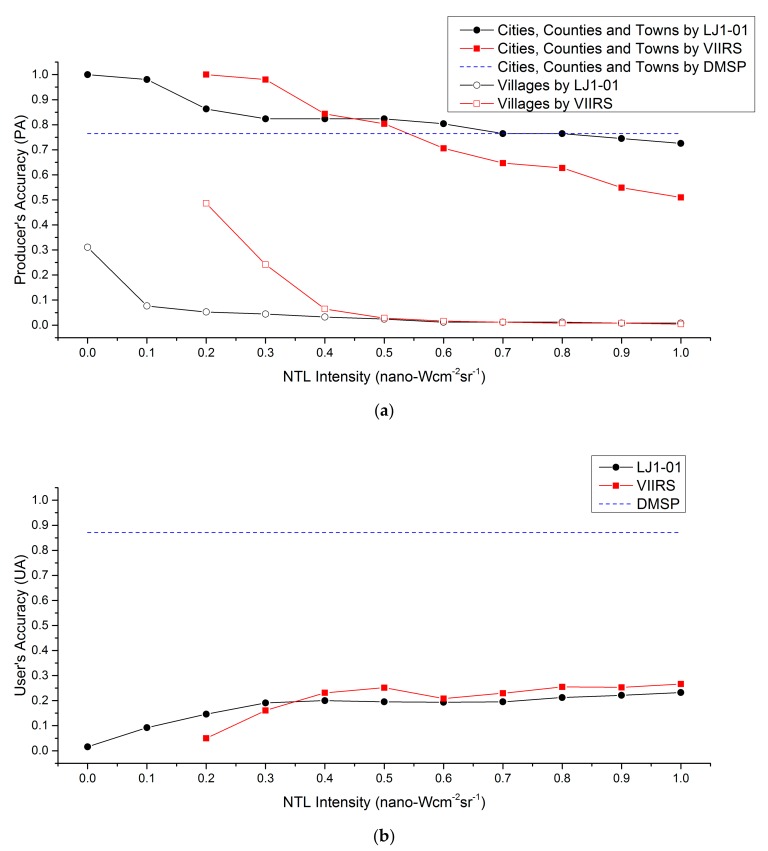
Detection accuracy of different NTL images: (**a**) producer’s accuracy of village-level and above village-level samples; (**b**) user’s accuracy of all samples. The detection accuracy of the DMSP image is calculated using the original image, and there is no cutoff threshold set for the DMSP image. Since no the village-level sample is detected by the DMSP image, the producer’s accuracy of the DMSP image for village-level sample is none. The footprints, which are smaller than four pixels in LJ1-01 NTL image, are eliminated when calculating user’s accuracy.

**Figure 8 sensors-19-03708-f008:**
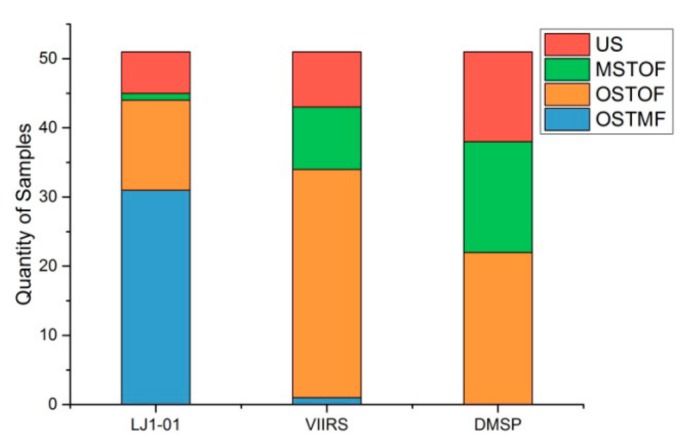
Spatial correspondence of different NTL images. US: Unidentified Samples, MSTOF: Multi Samples to One Footprint, OSTOF: One Sample to One Footprint, OSTMF: One Sample to Multi Footprints.

**Figure 9 sensors-19-03708-f009:**
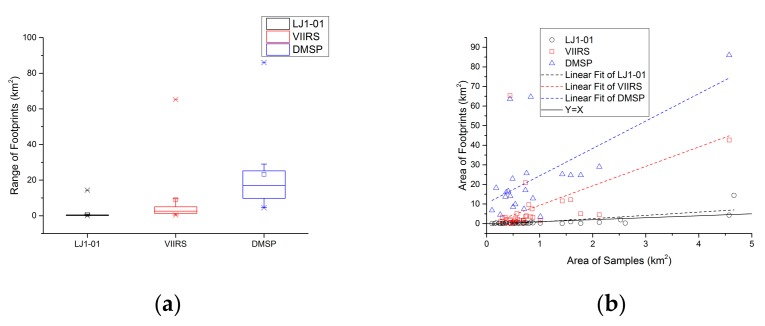
Area comparison between footprints and town-level samples: (**a**) The area statistics of town-level NTL footprints; (**b**) The area comparison with town-level NTL footprints and town-level samples.

**Figure 10 sensors-19-03708-f010:**
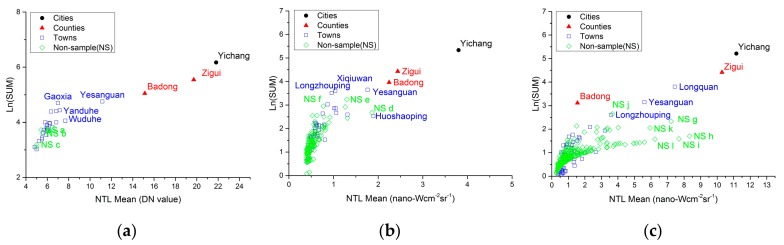
Total NTL intensity of footprints using different NTL data sources. The x-axis represents mean of footprints’ NTL intensity. The y-axis represents the logarithm of the total sum of the NTL: (**a**) Footprints of DMSP image; (**b**) Footprints of VIIRS image; (**c**) Footprints of LJ1-01 image. NS: non-sample, and the marks of the non-sample are consistent with that in Figure 11.

**Figure 11 sensors-19-03708-f011:**
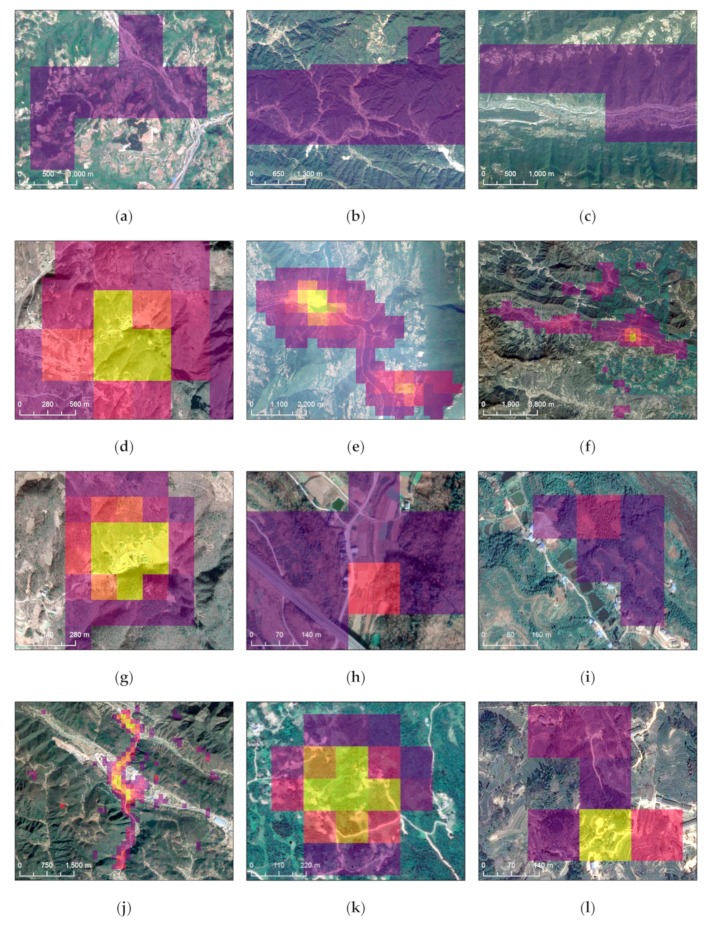
Validation of non-sample footprints with high-resolution images: (**a**–**c**) NTL’s footprints on DMSP image that may be caused by traffic and temporary engineering activities; (**d**–**f**) NTL’s footprints on VIIRS image that may be caused by industry and traffic; (**g**–**l**) NTL’s footprints on LJ1-01 image that may be caused by industry, traffic and villages.

**Figure 12 sensors-19-03708-f012:**
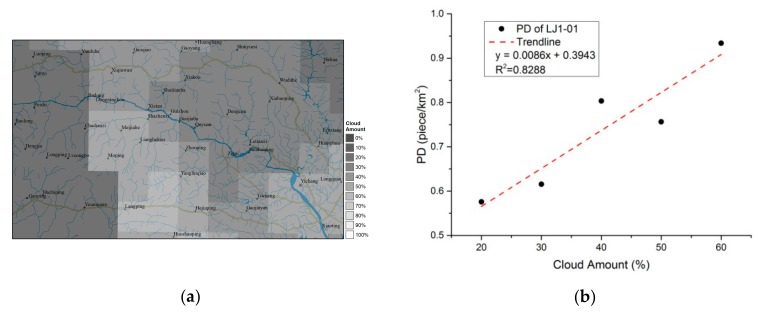
Clouds map and correlation between the number of footprints and the amount of cloud. (**a**) The mosaic cloud map on August 22, 2018 and September 3, 2018 from 22:00 to 23:00; (**b**) the correlation between the number of footprints and the amount of cloud in the study area.

**Figure 13 sensors-19-03708-f013:**
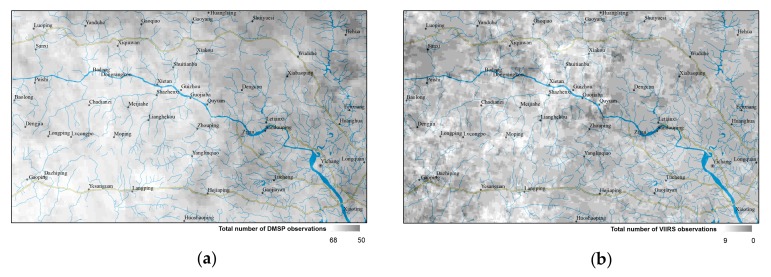
Number of composites. (**a**)Annual composites from DMSP; (**b**) Monthly composites from VIIRS.

**Table 1 sensors-19-03708-t001:** Parameters of the various NTL satellites.

Parameter	Satellite		
	DMSP/OLS	NPP/VIIRS	LJ1-01
Orbit height	850 km	827 km	645 km
Regression cycle	12 h	12 h	3–5 days
Spectral range	500–900 nm	500-900 nm	460–980 nm
Quantization bits	6	14	15
Spatial Resolution	2.7 km	740 m	130 m
Swath	3000 km	3000 km	264k m
On-board calibration	No	Yes	Yes
Nighttime overpass	~19:30	~1:30	~22:30
Available Period	1992–2013	November 2011–present	June 2018–present

**Table 2 sensors-19-03708-t002:** Landscape indices of NTL footprints.

Indicator	Calculation	Significance
Number of patches, NP	Total number of patches of NTL’s footprints	NP reflects the spatial pattern and heterogeneity of NTL’s footprints; it is positively correlated with fragmentation of NTL’s footprints.
Class area, CA	Total class area of NTL’s footprints	CA determines the range of NTL’s footprints; It indicates the scope of human activity.
Patch density, PD	The NP/CA ratio of NTL’s footprints	PD is the number of patches per unit area; it reflects the dispersion of footprints of NTL.
Mean patch size, MPS	The CA/NP ratio of NTL’s footprints	MPS indicates the fragmentation of NTL’s footprints; it also reflects NTL’s footprints heterogeneity.

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
