# Peer review of "Assessing the Ability of Luojia 1-01 Imagery to Detect Feeble Nighttime Lights"

_sensors, 2019, doi:10.3390/s19173708_

Round 1
Reviewer 1 Report
Comments for sensors
This paper evaluated the potential of LJ1-01 to detect feeble nighttime lights by comparing with DMSP and VIIRS data. This study is very interesting, but some problems still need to be addressed.
(1) The literature review in the Introduction is insufficient, especially for the recent researches on LJ1-01 nighttime lights. For example,
Ou, J., Liu, X., Liu, P., Liu, X., 2019. Evaluation of Luojia 1-01 nighttime light imagery for impervious surface detection: A comparison with NPP-VIIRS nighttime light data. Int. J. Appl. Earth Obs. 81, 1-12.
Jiang, W. et al., 2018. Potentiality of Using Luojia 1-01 Nighttime Light Imagery to Investigate Artificial Light Pollution. Sensors, 18(9): 2900.
(2) Do the samples indicate human settlement in the study? How do they obtained from the Google Map image? Please provide detailed description about the samples.
(3) In Line 223-224, authors stated that the optimal cutoff threshold for VIIRS was set to 0.4 and that for LJ1-01 to 0.2 according to the above analysis. However, from Figure 7, it seems difficult to see that detection accuracy and detection efficiency had an optimal value when the cutoff threshold for VIIRS was 0.4 and that for LJ1-01 was 0.2.
Author Response
Response to Reviewer 1 Comments
I would like to thank you all for your efforts in reviewing my paper. It is your comments that make this paper technically and literally better. Besides, we hope that based on this work, some other meaningful researches can be conducted in the future. In terms of the comments, there are some main revisions of this paper as below:
Point 1: The literature review in the Introduction is insufficient, especially for the recent researches on LJ1-01 nighttime lights. For example,
Ou, J., Liu, X., Liu, P., Liu, X., 2019. Evaluation of Luojia 1-01 nighttime light imagery for impervious surface detection: A comparison with NPP-VIIRS nighttime light data. Int. J. Appl. Earth Obs. 81, 1-12.
Jiang, W. et al., 2018. Potentiality of Using Luojia 1-01 Nighttime Light Imagery to Investigate Artificial Light Pollution. Sensors, 18(9): 2900.
Response 1:
Thank you so much for your suggestion. We have added the latest researches of LJ 1-01, including “urban mapping [26], estimating socio-economic parameters [27] and urban community housing price [28], impervious surface detection [29] and investigating artificial light pollution [30]”. We put these supplementary materials in the fourth paragraph of Introduction.
Point 2: Do the samples indicate human settlement in the study? How do they obtained from the Google Map image? Please provide detailed description about the samples.
Response 2:
The samples at each level were randomly selected from the National Geodatabase. Since the human settlements in the database are point-vector data, it is necessary to manually extract the urbanized area of each sample according to the high-resolution images from Google Map.
The sizes of urbanized area of different human settlement levels may be variable around the world. In the study area of this paper, the size of urbanized area of the city-level samples is greater than 150km2. The average sizes of urbanized area of the county-level and town-level samples are 26.7km2 and 0.87km2 respectively. Then, the average size of urbanized area of the village-level samples is smaller than 0.1km2.
Point 3: In Line 223-224, authors stated that the optimal cutoff threshold for VIIRS was set to 0.4 and that for LJ1-01 to 0.2 according to the above analysis. However, from Figure 7, it seems difficult to see that detection accuracy and detection efficiency had an optimal value when the cutoff threshold for VIIRS was 0.4 and that for LJ1-01 was 0.2.
Response 3:
There is no doubt that cutoff threshold setting is very important. A low cutoff threshold cannot effectively reduce noise while a high cutoff threshold may filter out too much weak NTL information. Unfortunately, it is difficult to accurately determine the optimal cutoff threshold using a quantitative approach. However, by analyzing Figure 6 and Figure 7, the range of the optimal threshold can be roughly determined by manual assistance. Figures 6c and 6d suggest that the optimal threshold for LJ1-01 should be greater than 0.1, while the optimal threshold for VIIRS should be greater than 0.3 or 0.4. Figure 7a illustrates that the producer accuracy are relatively stable while the LJ1-01 cutoff threshold is between 0.2 and 0.6, and the VIIRS cutoff threshold is between 0.4 and 0.5. Figure 7b shows that when the LJ1-01 cutoff threshold is greater than 0.3 and the VIIRS cutoff threshold is greater than 0.4, the user’s accuracy are relatively stable, but there is not much difference from the previous one. According to the above analysis, the optimal cutoff threshold for VIIRS was set to 0.4 and that for LJ1-01 to 0.2. Based on the above steps, we have revised the relevant content in Section 3.3.
|
(a) |
(b) |
|
(c) |
(d) |
Figure 6. Landscape indices of footprints in different VIIRS and LJ1-01 NTL images: (a) NP; (b) CA; (c) PD; (d) MPS.
|
(a) |
|
(b) |
Figure 7. Detection accuracy of different NTL images: (a) producer’s accuracy of village-level and above village-level samples; (b) user’s accuracy of all samples. The detection accuracy of the DMSP image is calculated using the original image, and the cutoff threshold is not set for the DMSP image. Since no the village-level sample is detected by the DMSP image, the producer’s accuracy of the DMSP image for village-level sample is none. The footprints, which are smaller than four pixels in LJ1-01 NTL image, are eliminated when calculating user’s accuracy.

Reviewer 2 Report
The authors compared the suitability of three different types of nighttime light (NTL) images in capturing weakly lit areas such as towns and villages. While knowing the capability of NTL in detecting areas with sparse human activities is important, this study suffers from several issues. The results are obvious even without any investigation because the new sensor has several advantages over OLS and VIIRS.
(1) the motivation of this study is not well stated in the Introduction section.
(2) The methodology is not clear, especially for the determination of the optimal thresholds. Without this clarification, readers may get confused about how they are determined and their impacts on the detection accuracy of towns and villages. For example, how those thresholds were determined in Line 224, Page 8. Figure 8 will be different if you change these values. In addition, the authors should state the reasons for keeping the threshold of DMSP as 4.
(3) What is your definition of weekly lit areas, villages, towns, or counties? The X-axis of Figure 5, 6, and 7 ranges from 0 to 1.0. It seems that the authors only considered cutoff values between this range, the reasons of which should be addressed (related to the definition).
(4) The authors thought the Three Gorges Reservoir region is the “suitable area” (as stated in the objectives) to compare NTL images. However, using a single study area, it is hard to make a quantitative conclusion regarding how much the improvement of the new sensor to the others in identifying towns and villages. Thus, I recommend the authors should select more study areas across the country or global to get more quantitative results.
(5) The authors only tested one aspect of the detection accuracy for different NTL images (i.e., samples that are detected by NTL, producer’s accuracy). However, the authors should also measure the user’s accuracy of NTL-derived patches because the LJ image provided more clusters (some are noise instead of towns and villages) and it is not convincing to indicate that LJ is better than VIIRS and DMSP/OLS without information of user’s accuracy.
Reviewer 3 Report
The paper compares Luojia 1-01 (LJ1-01) with VIIRS and DMSP images products to Detect Feeble Nighttime Lights. It contributes to disseminate NTL uses, and especially to present potential and limitations of NTL sensors of finer spatial resolution, as LJ1 provides.
In order to publish in Sensors, some concepts and procedures need to be clarified in the text, as well as some details:
1) The title may bring the idea that the study site is relevant for the discussion, as for the use of NTL for land-use monitoring. However, the central idea of the article is the LJ1 potential for low-intensity NTL detection. The title should be adjusted.
2) The paper compares three different products: LJ1, VIIRS, and DMSP. They have different spatial resolution, and temporal coverage. Instead of putting LJ1 as the most efficient, it is necessary to keep in mind that they are suitable for different applications, and it must be clear throughout the text. It is not “fair” to expect that an annual DMSP product would be able to detect “feeble” NTL, as a daily LJ1 product would do.
3) “ feeble” NTL - this is not a usual term and should be defined in the text. Also, why it is important to detect this type of NTL? Some references would reinforce your explanation. Also, how can you distinct feeble NTL from noise, or non-stable NTL???
4) Some details about the study site would enable better results interpretation for a broader audience. Especially about the size of urbanized areas, that is very variable around the world – a city, town, village in China could be very different than in Europe or the US.
5) LJ1 mosaic image seems to have a difference in NTL patches detection, comparing the west vs east part image of the study site (Fig 3c). How is the images articulation? (84 – “To cover the entire study area, two images acquired on August 18, 2018 and September 3, 2018 were obtained”).
6) Landscape indices. Using landscape metrics is interesting for remote sensing approach, but they are usually selected/built considering the ecological issue that will be treated. The cited reference (393- 17 – Zang et al. 2017) deals with ecosystem services subject, and so, it is not adequate to justify what you are proposing these metrics. Consider justifying better your choices, with specific literature support.
7) Table2. What is “the specific landscape type”?
8) 138 – include a description of NP for Equation (4)
9) 157 – Figure 6b?
10) Correlation between histograms curves?? To compare frequency distribution (as in a histogram), you should use another type of statistic test. Correlation is not applicable. Consider better explain the histogram comparison analyze.
11) 173 – Fig 6 - the graphics do not show a linear decrease. Rephrase.
12) 179 - Fig 6 – improve the legend text, explaining that X-axis is related to cutoff threshold values.
13) Noise x feeble – this is central in the article. For users, it is very important to have a good analysis and/or indication of procedures to detect and avoid noise in LJ1 images.
14) 186 – the average size (area and population) of the settlements would improve your results explanation
15) 186 – what is the purpose of “which is why such samples are less frequently selected” ??
16) Fig7 – If " the digital number (DN) was used to represent relative brightness" for DMSP. What is plotted in this Figure? x-axis = NTL intensity (nano Wcm-2sr-1) ??
17) 227 – include also the legend (MSTOF, OSTOF, OSTMF) in the text.
18) What is a halo? this is not a usual term and should be explained in the text (Is it the overglow?) How does it happen in LJ1, VIIRS, and DMSP? And why?
19) 243 – I could not appreciate “the highest consistency” by Fig 9. Maybe, the difference (in area) between the estimated (NTL) and observed (samples) could better justify this point.
20) Fig 10 – improve figure resolution and size. What are the units in x-axis?
21) 276 – What are temporary engineering activities? Be more specific with examples.
22) 282 – Fig 11 - It would be better to keep the same scale of Google data for each sensor. And, Explain the NTL source for each case. (General readers are not familiar to the Chinese geography/land use cover types).
23) 290 – How did you detect noise? How users should proceed to eliminate noise in LJ1?
24) 292 – Moon and Cloud information “can be obtained” OR did you use this data to check this information here?
25) Fig 12 – The moon phases are beautiful, but pointless here. Just keep the information in the text, and consider bringing the graphic of Fig13 to Fig12.
26) 329 – the discussion about overpass time. Put it in relative perspective: at 22:30 there are more influence of lights from traffic - so, NTL will detect human activity, but it would be combined with lights from urban infrastructure (that is not good if you are interested in intra-urban zones). It can be valuable, depending on the objective.
27) 338- consider review the previous text to explain and give a precise definition for the terms: “weak NTL”; “Weak human activity detection”; “weak information on human activities”.
28) “In addition, the overpass time of the LJ1-01 also makes its data more valuable.” à see the previous comment.
29) Conclusions: again – instead of trying to magnify LJ1 sensor comparing to VIIRS and DMSP (it is obvious that the LJ1 spatial resolution will guarantee that “ LJ1-01 NTL imagery has a better potential for NTL detection”), just bring the relevant information that you describe for LJ1, emphasizing that it enables a complementary data for finer scales. What were the smallest settlements detected by LJ1 (in area or population)?? How much LJ1 patches were not stable lights (of what size distribution)?
The pdf file (attached) has some comments marked.

Round 2
Reviewer 2 Report
The authors have answered all my questions, although I do not think this study is novel enough. More seriously, the conclusions contribute very little to existing research given that the study is limited to a single site (the title may indicate the authors are trying to assess the capability at broader scale - country, continental, or global) . The conclusions in the abstract can be obtained even without this study.
Reviewer 3 Report
The authors managed to answer the majority of comment/question/suggestion that I have previously pointed out in the manuscript. This has greatly improved the text.
There are still some details (present below), and my last concern is about the writing style which contains some informality. A style review would better fit the document to the magazine's standards.(For example 206. "Now let's go back to the noisy situation").
Important POINT:
Point 5:
Response 5:
The study area is located at the junction of the two LJ1-01 images. Since there is an overlapping strip between the two images, the number of NTL footprints in the strip appears to be more than anywhere else (shown as the following figure). In section 4.2, the relationship between the number of NTL patches and the intensity of moon’s light and the amount of clouds is represented.
- Ok, you´ve discussed the cloud effect, but not the strip overlapping. This explanation has to be on the text.
Details:
88." therefore the local population density is low".- how low? quantify it.
281. higherlevel samples
309. (a) the tile axis is missing. There is a "." instead
313. NS - should be in the figure legend.
Round 3
Reviewer 2 Report
I totally understand the authors' statement that pixels with low NTL intensity are informative and useful. However, this study is not convincing enough to readers in terms of research design, method description, and the usage of these conclusions. I keep rejecting this manuscript because this paper lacks that depth in either method development or investigating the implications of the findings. If the authors want to focus on method development (e.g., identifying villages and rural settlements), it should be reviewed in the Introduction section, fully stated in the methodology section, and assessed in the results and discussion section. More importantly, the implications of this study should be further stated because these findings may vary across study areas. For example, villages and counties in the United States are better lit at night compared to Chinese ones. I really hope the authors can consider these. At this stage, I will recommend accepting this version of manuscript because it seems the authors do not want to redesign their methodology and add more test sites, but major language edits are required, which is important to make this manuscript more readable.
Author Response
English language and style of the paper are extensive edited. Your considerate understanding, and acceptance of the study would be greatly appreciated. Your review has played an important role in improving the quality of this article.